# Assessment of Mastectomy Skin Flaps for Immediate Reconstruction with Implants via Thermal Imaging—A Suitable, Personalized Approach?

**DOI:** 10.3390/jpm12050740

**Published:** 2022-05-01

**Authors:** Hanna Luze, Sebastian Philipp Nischwitz, Paul Wurzer, Raimund Winter, Stephan Spendel, Lars-Peter Kamolz, Vesna Bjelic-Radisic

**Affiliations:** 1Division of Plastic, Aesthetic and Reconstructive Surgery, Department of Surgery, Medical University of Graz, 8036 Graz, Austria; sebastian.nischwitz@medunigraz.at (S.P.N.); office@wurzer-medical.at (P.W.); r.winter@medunigraz.at (R.W.); stephan.spendel@medunigraz.at (S.S.); lars.kamolz@medunigraz.at (L.-P.K.); 2Burgenländische Krankenanstalten-Ges.m.b.H., 7000 Eisenstadt, Austria; 3COREMED–Cooperative Centre for Regenerative Medicine, JOANNEUM RESEARCH Forschungsgesellschaft mbH, 8010 Graz, Austria; 4Research Unit for Safety and Sustainability in Healthcare, Division of Plastic, Aesthetic and Reconstructive Surgery, Department of Surgery, Medical University of Graz, 8036 Graz, Austria; 5Breast Unit, Helios University Hospital, University of Witten Herdecke, 42283 Wuppertal, Germany; vesna.bjelic-radisic@helios-gesundheit.de; 6Division of General Gynaecology, Department of Obstetrics and Gynaecology, Medical University of Graz, 8010 Graz, Austria

**Keywords:** personalized medicine, thermal imaging, reconstructive breast surgery, mastectomy-skin-flap perfusion, mastectomy-skin-flap necrosis

## Abstract

Background: Impaired perfusion of the remaining skin flap after subcutaneous mastectomy can cause wound-healing disorders and consecutive necrosis. Personalized intraoperative imaging, possibly performed via the FLIR ONE thermal-imaging device, may assist in flap assessment and detect areas at risk for postoperative complications. Methods: Fifteen female patients undergoing elective subcutaneous mastectomy and immediate breast reconstruction with implants were enrolled. Pre-, intra- and postoperative thermal imaging was performed via FLIR ONE. Potential patient-, surgery- and environment-related risk factors were acquired and correlated with the occurrence of postoperative complications. Results: Wound-healing disorders and mastectomy-skin-flap necrosis occurred in 26.7%, whereby areas expressing intraoperative temperatures less than 26 °C were mainly affected. These complications were associated with a statistically significantly higher BMI, longer surgery duration, lower body and room temperature and a trend towards larger implant sizes. Conclusion: Impaired skin-flap perfusion may be multifactorially conditioned. Preoperative screening for risk factors and intraoperative skin-perfusion assessment via FLIR ONE thermal-imaging device is recommendable to reduce postoperative complications. Intraoperative detectable areas with a temperature of lower than 26 °C are highly likely to develop mastectomy-skin-flap necrosis and early detection allows individual treatment concept adaption, ultimately improving the patient’s outcome.

## 1. Introduction

Breast cancer (BC) ranks top in malignancies among females worldwide and represents the most common reason for death by cancer as well [1]. About two-thirds of patients with BC undergo breast-conserving surgery [2,3]. Thirty to forty per cent require a mastectomy and about 25% of these patients decide to undergo an immediate breast reconstruction (IBR) [2,3]. Besides autologous tissue reconstruction, IBR with implants is a commonly used technique and considered safe from an oncological point of view [2]. Larger varieties of sizes and shapes of implants and the use of different meshes creating a more natural appearing lower pole, may have contributed to an increased use of implants [4,5].

Due to the combination of oncoplastic and plastic surgical techniques, aesthetically pleasing reconstructive results can be achieved without compromising the oncological safety [6]. A modified radical mastectomy was replaced, e.g., by nipple-sparing mastectomy (NSM) and/or skin-sparing mastectomy (SSM), offering good aesthetic results [4].

Besides hematoma and infection listed as the most common early complications in IBR with implants, wound-healing disorders (WHD) and mastectomy-skin-flap necrosis (MSFN) occur more commonly than appreciated, with reports ranging from 5% to 41.2% [7,8,9,10,11]. The main reason for WHD and especially for skin-flap necrosis is attributed to an impaired perfusion of the remaining skin flap after mastectomy [11]. Subsequently, numerous challenging sequelae including wound-management problems, follow-up interventions, implant loss, delays to adjuvant therapy, aesthetic compromise and patient dissatisfaction can occur [11].

Several studies have investigated patient-related (e.g., smoking, diabetes, obesity etc.) and surgery-related risk factors (e.g., incision type, mastectomy weight, thickness of the skin flap, etc.) for WHD and MSFN to date [11,12]. Apart from screening and monitoring possible risk factors, intraoperative assessment of the individual skin-flap perfusion is considered of the utmost importance to detect areas at risk of WHD and MSFN [12]. A number of assessment methods have evolved so far, including clinical evaluation, handheld Doppler devices, laser Doppler, fluorescein angiography and indocyanine green techniques [12]. As an indicator of tissue perfusion, skin (surface) temperature can also readily and accurately be measured via thermal imaging [13,14]. The present study evaluated the feasibility of assessing the individual mastectomy-skin-flap perfusion via the thermal-imaging device FLIR ONE in patients undergoing NSM or SSM and following IBR with implants. Furthermore, possible patient-, surgery- and environment-related risk factors for postoperative complications were determined and the overall complication rate was assessed.

## 2. Methods

This prospective analysis was conducted the Medical University of Graz, Austria, Division of General Gynaecology, Department of Obstetrics and Gynaecology and Division of Plastic, Aesthetic and Reconstructive Surgery, Department of Surgery in cooperation with the Department of Surgery, LKH Graz II, Standort West, Graz, Austria between 2016 and 2018 and has been approved by the responsible ethics’ committee. The surgeries were performed by two different surgeons (one female and one male) who were experienced in this field for 28 and 31 years, respectively.

### 2.1. Study Population

Fifteen female (age range 18–80 years) healthy, non-pregnant study participants were enrolled. Inclusion criteria were defined as follows: history of BC, carcinoma in situ on one or both breasts and/or known genetic BRCA I or BRCA II mutation and planned NSM or SSM, and following IBR with definite implants. Exclusion criteria were defined as follows: BC diagnosis with the contraindication for NSM or SSM (e.g., inflammatory carcinoma), use of tissue expanders; diabetes mellitus type 1 and 2, nicotine abuse, inability to fully comprehend study procedures or to provide written informed consent. The reconstruction was performed in dual plane technique using anatomical silicone implants (Mentor Deutschland GmbH, Munich, Germany), placed subpectorally. The acellular dermal matrix Strattice™ (Allergan, Dublin, Ireland) was used for implant stabilization and affixed to the musculus pectoralis major as well as in the inframammary fold. Postoperative follow-up visits were uniformly scheduled at 1, 2 and 6 weeks postoperative.

Additional patient-related, (age, Body-Mass-Index (BMI), body temperature), surgery-related (incision type and position, implant size, surgery duration) and environment-related data (room temperature) were acquired.

### 2.2. FLIR ONE

Surface temperature was acquired using measurements obtained with a FLIR ONE thermal-imaging camera (FLIR Systems, Wilsonville, OR, USA). FLIR ONE is a lightweight, pocket-sized, smartphone attachment thermal-imaging camera with a measurable temperature range from −20 °C to 120 °C and a measurement accuracy of 0.10 °C [15]. A Multi-Spectral Dynamic imaging technology allows for enhanced thermal imaging by embossing details from the camera onto the thermal image [15].

Four timepoints for thermal imaging with FLIR ONE were determined:-Preoperative: immediately after anesthesia induction before disinfection-Intraoperative 1: immediately after NSM/SSM-Intraoperative 2: immediately after implant placement and wound closure-Postoperative: 24 h postoperative

The distance between the FLIR ONE and patients’ skin was set at 30 cm in every measuring. At recording, surgical lights were turned off and body and room temperature were assessed. Thermal images were transferred to FLIR Tools software, where highest, lowest and average temperature of each image was determined within a specialized region of interest (ROI). (See Figure 1) The ROI was manually plotted, matching the region, where the NSM/SSM was performed.

### 2.3. Statistical Analysis

Since this study was designed as an explorative study with a small sample size, a formal sample size calculation was waived. The rationale for conducting a study with females exclusively was based on the fact that NSM/SSM with IBR with implants is primarily used in female BC patients. Data were analyzed with GraphPad Prism software (version 9.0.2; GraphPad Software, Inc., San Diego, CA, USA). Mean, median and standard deviation (SD) of numerous data were calculated and correlated to the occurrence of postoperative complications (WHD, MSFN) performing a Spearman correlation test. All statistical tests were two-tailed and differences were considered statistically significant when *p* < 0.05.

## 3. Results

Overall, 15 patients with a mean age of 44.1 years (SD ± 9.2 years) and a mean BMI of 25.9 kg/m^2^ (SD ± 2.46 kg/m^2^) were included in this investigation. All patients enrolled underwent neoadjuvant chemotherapy; no patient underwent radiotherapy prior to surgery. Risk-reduction mastectomy was performed in one, NSM in six and SSM in seven patients. In the patient undergoing risk-reduction mastectomy, skin-flap perfusion was assessed exclusively on the left side, since a minimally prefilled tissue expander was used on the right side. The mean surgery duration was 2 h 32 min (SD ± 108 min). 8 (53.33%) periareolar and 7 (46.67%) inframammary incisions were performed.

### 3.1. Surface Temperature of the Mastectomy Skin Flap

The preoperative mean surface temperature was 36.3 °C (SD ± 3.6 °C). The first mean intraoperative surface temperature after subcutaneous mastectomy (intraoperative 1, I1), was 33.3 °C (SD ± 4.0 °C) decreasing to 32.0 °C (SD ± 2.7 °C) and at the second timepoint after implant placement (intraoperative 2, I2). Twenty-four hours postoperative, the mean surface temperature was 36.0 °C (SD ± 1.4 °C). The preoperative mean temperature was statistically significantly higher than intraoperative 1 and intraoperative 2 (*p* = < 0.001). Intraoperative measured mean temperatures were statistically significantly lower than postoperative (I1: *p* = 0.024, I2: *p* = 0.017).

The highest mean preoperative temperature was 37.5 °C (SD ± 1.1 °C). intraoperative measured mean highest temperature statistically significantly decreased to 35.5 °C (SD ± 1.3 °C) at the first timepoint and 34.6 °C (SD ± 1.9 °C) at the second (p= 0.001 and *p* = 0.048). Twenty-four hours postoperative, the mean highest temperature was 37.4 °C (SD ± 1.2 °C), which was statistically significantly higher than both values measured intraoperatively (*p* ≤ 0.001 and *p* = 0.002).

The lowest preoperative temperature was 35.5 °C (SD ± 3.0 °C). The intraoperative measured lowest temperature statistically significantly decreased to 28.3 °C (SD ± 4.4 °C) after subcutaneous mastectomy and to 27.7 °C (SD ± 4.6 °C) after implant placement (*p* ≤ 0.001 and *p* = 0.036) in the same patients. 24 h postoperative, mean lowest temperature was 33.7 °C (SD ± 3.0 °C), which was statistically significantly higher than both values measured intraoperatively (*p =* 0.012). An overview of surface temperatures is depicted in Table 1 and Figure 2. 

### 3.2. Wound-Healing Disorders and Necrosis

Patients were divided into a “no-complication group” and a “complication group” including WHD and MSFN for further comparisons. 4 of 15 patients (26.7%) developed WHD, which were initially treated without surgical interaction but prolonged dressing changes and administration of antibiotics. In the further course, 3 (20%) WHD converted into superficial MSFN requiring follow up interventions under local anesthesia. No implant loss was noted. In all patients of the complication group, WHD and MSFN occurred within areas with an intraoperative measured temperature lower than 26 °C.

The main statistically significant difference between the no-complication and the complication group was found regarding the surface temperatures of the mastectomy skin flap. The complication group showed a statistically significantly lower mean temperature in preoperative (*p* ≤ 0.001), intraoperative 1 (*p* = 0.029) and postoperative (*p* ≤ 0.001) measurements. The mean lowest temperatures were statistically significantly lower during the entire procedure (*p* ≤ 0.001) in the complication group, as well as the mean highest temperatures pre- and postoperative (*p* = 0.021 and *p* ≤ 0.001). An overview of surface temperatures within the no-complication and complication groups is depicted in Table 2.

#### 3.2.1. Surgery-Related Risk Factors

WHD rate with periareolar incisions was 37.5% vs. 14.3% with inframammary incisions. MSFN rate, requiring follow-up interventions under local anesthesia in all patients, was 25% with periareolar incisions and 14.3% with inframammary incisions. No statistically significant difference between incision type (*p* = 0.36) and NSM or SSM technique (*p* = 0.29) was observed. An overview of incision types and postoperative complications is depicted in Figure 3.

A comparison of both groups showed a statistically higher average surgery duration in the complication group (2 h 58 min) compared to the no-complication group (2 h 22 min) (*p* = 0.027). Furthermore, statistically significantly lower average room temperatures (21.2 °C (SD ± 0.2 °C), *p* ≤ 0.001) were observed in the complication group in comparison to the no-complication group (23.0 °C (SD ± 1.2 °C)). In all patients, mastectomy weight approximately correlated to the implant size. A trend towards larger implants was noted in the complication group (370 cm^3^ (SD ± 43.3 cm^3^)) compared to the no-complication group (320.5 cm^3^ (±42.5 cm^3^), *p =* 0.223).

#### 3.2.2. Patient-Related Risk Factors

Statistically significantly lower mean body temperatures during the entire procedure (35.6 °C (SD ± 0.4 °C), *p* = 0.012) were found in the complication group when compared to the no-complication group (36.0 °C (SD ± 0.1 °C). Patients developing WHD and MSFN also had a significantly higher BMI (28.1 kg/m^2^ (SD ± 0.6 kg/m^2^)) than patients without complications (25.3 kg/m^2^ (SD ± 3.0 kg/m^2^), *p* = 0.002). No age-related significances were found (*p* = 0.46). Demographic and clinical data of the complication and no-complication groups are listed in Table 3.

## 4. Discussion

Wound-healing disorders and mastectomy-skin-flap necrosis of the remaining skin flap after mastectomy due to hypoperfusion are highly relevant and underappreciated complications that may result in considerable challenges for the patient and health-care system. Potential consequences range from aesthetic compromise and patient dissatisfaction to limited options of reconstruction and delays in adjuvant therapies [10]. Existing evidence highlights the difficulty in assessing individual mastectomy-skin-flap perfusion despite various techniques available to date [16].

### 4.1. Examination of Skin-Flap Perfusion via Thermal Imaging

The present study demonstrated thermal imaging via FLIR ONE to be a suitable approach for the measurement of individual skin/surface temperature as a proxy indicator of tissue perfusion. The FLIR ONE particularly excels in the domains of usability, time to image acquisition, and reliably accurate results requiring minimal to no training, resulting in high-resolution images. In our opinion, these features contribute to the fact that the FLIR ONE can be considered a valuable support tool in clinics in a wider range of applications. Personalized thermal imaging may be of particular value in establishing individual treatment concepts. To date, thermal imaging via FLIR ONE has widely been clinically employed, for instance, in the individual assessment of burn wounds [13,17,18,19], diagnosis of complex regional pain syndrome [20], detection of perforator vessels in reconstructive surgery [14,21,22] and many other settings where thermal distribution patterns can yield proxy data. Within the field of breast oncology in particular, thermal imaging has recently been investigated as an emerging modality in breast-cancer screening [23,24] and has been considered a helpful tool in the treatment of breast-cancer-related lymphedema [25]. Apart from the low initial costs and ease of use of the FLIR ONE device, the non-invasive, no-touch character and absence of radiation are among the many advantages of this technique [22,23].

### 4.2. Detection of Hypoperfused Areas at Risk for WHD and MSFN

The procedure of mastectomy interrupts the axial perfusion of the over-lying breast dermis, leaving the relatively hypovascular skin flaps dependent on random-pattern perfusion and drainage through the subdermal plexus [26]. Studies report mastectomy flaps with a higher amount of subcutaneous fat and, therefore, better preserved blood supply, to be associated with a reduced risk of MSFN [26,27].

Evidence indicates a direct relationship between changes in tissue perfusion and temperature, therefore suggesting non-invasive surface-temperature measurement a valuable proxy marker for the analysis of (skin) perfusion [28]. In the present project, the temperature profile assessed via thermal imaging demonstrated a statistically significant drop during, and a rise after the procedure, nearly reaching initial values 24 h postoperative. The temperature drop is mainly induced due to the procedure of mastectomy and keeps increasing with increasing surgery duration, verifying the interruption the axial perfusion during the procedure. The subsequent postoperative increase may be attributed to a compensatory increase in random pattern perfusion and drainage through the subdermal plexus as well as room temperature compared to the operating room.

A comparison between patients without complications and those developing WHD and MSFN revealed significant differences in the surface temperature. We encountered significantly lower average temperatures in three out of four measuring time points, (*p* ≤ 0.001, 0.029 and <0.001); and lower maximum temperatures prior to and following the procedure (*p* ≤ 0.001) in the complication group. The lowest temperatures were statistically significantly lower during the entire procedure, suggesting lower tissue perfusion.

The present study revealed that areas with intraoperative temperatures lower than 26 °C are highly likely to develop WHD and MSFN, since their occurrence was exclusively noted within these areas. We derive that consideration of the proxy parameter of intraoperative temperature distribution to assess for clinically relevant hypoperfusion may lead to opportunities for early intervention in selected cases to avoid or reduce some of the possible adverse consequences. Early intervention is particularly important in IBR with implants and may comprise different strategies. The importance of the excision of non-viable skin edges to prevent WHD has often been determined in acute and chronic wound management [29]. According to our findings, excision of wound edges with a temperature of lower than 26 °C may be considered even when clinical signs of hypoperfusion are absent. If larger areas manifest with impaired skin-flap perfusion as indicated on thermal imaging with areas <26 °C, the risk of extended MSFN may be increased and the reconstruction strategy should be reconsidered; however, to our knowledge, no studies addressing this challenge exist to date.

### 4.3. Detection of Patient- and Surgery-Related Risk Factors for WHD and MSFN

In the present study, a significantly higher BMI and a trend towards larger implants matching the mastectomy weight were identified to be associated with flap morbidity. The majority of the existing literature investigating risk factors related to WHD and MSFN is limited to retrospective series [12]. Smoking, diabetes, radiotherapy, previous scars and severe medical comorbidity have been revealed as patient-related risk factors so far [12]. Similar to our results, there is also evidence that obesity and increased breast volumes may cause or contribute to impaired skin-flap perfusion [30,31].

Preoperative screening for known risk factors and incorporation into operative planning, especially if immediate breast reconstruction is performed, is of utmost importance. Unfortunately, the majority of patient-related risk factors are not modifiable prior to surgery; therefore, surgical risk factors need to be minimized. Among surgical factors, longer surgery duration is considered a risk factor for postoperative complications [12]. Our results not only revealed a statistically significant correlation to surgery duration, but also to body and room temperature. Measurements obtained during the procedure demonstrated statistically significantly lower values in both body and room temperature, in patients developing WHD and MSFN. Therefore, intraoperative temperature monitoring and management, if necessary, may be another approach to reduce postoperative complications, especially when other risk factors are present. No differences were identified between inframammary and periareolar incisions; however, other authors attribute flap morbidity to wise pattern mastectomy incisions [31].

In our setting, patients developing MSFN only required one follow-up intervention under local anesthesia and no implant loss was noted. However, extended MSFN resulting from hypoperfused areas may consequently lead to a higher chance for subsequent implant loss. Woerdeman et al. further reported an increased risk of implant loss in patients with larger-than-average-sized breasts and obese smokers [30]. In these patients, placement of breast implants may particularly need to be refrained from. As an alternative, tissue expanders may be placed first in order to minimize pressure on the skin flap and prevent subsequent implant loss.

A combination of careful preoperative planning and intraoperative monitoring may contribute to a reduced incidence of WHD resulting in MSFN. Early intervention in selected cases and deviation from the planned reconstructive procedure may reduce the overall morbidity of MSFN.

### 4.4. Limitations

This study is limited due to the small sample size, hence results and patient-related factors in particular may not be fully representative and transferable. Despite some significances that were statistically demonstrated, the small sample size might be accountable for the lack of further significances; however, we believe that tendencies were established to guide further investigation. While the study setup was designed to reduce possible bias caused by different surgeons, the thickness of the mastectomy skin flap linked to the skin-flap perfusion may not be directly comparable in all patients.

Furthermore, previous breast surgeries were not evaluated. Consequently, a bias by individual factors—such as scars potentially impairing the remaining skin flap perfusion—must be considered as a limiting factor. However, evaluation of a possible influence of previous medical interventions on skin-flap perfusion and postoperative complications may be an interesting approach for future studies. Ultimately, there is a great demand for strategies to detect risk factors contributing to WHD and MSFN. Further studies are necessary to reach the full potential of thermal imaging in the detection of areas at particular risk for developing WHD and MSFN. 

## 5. Conclusions

WHD and MSFN due to compromised perfusion patterns of the remaining skin flap occur more commonly than appreciated, leading to numerous challenges. Hypoperfusion of the remaining mastectomy skin flap as a major factor contributing to WHD and MSFN is readily and accurately examined via the novel and personalized approach of thermal imaging with the FLIR device. Skin areas with a temperature lower than 26 °C are highly likely to develop subsequent WHD and MSFN and may require early intervention to avoid or reduce the incidence of MSFN. Ultimately, careful and diligent individual preoperative planning and intraoperative monitoring may contribute to a reduced incidence of WHD converting to MSFN.

## Figures and Tables

**Figure 1 jpm-12-00740-f001:**
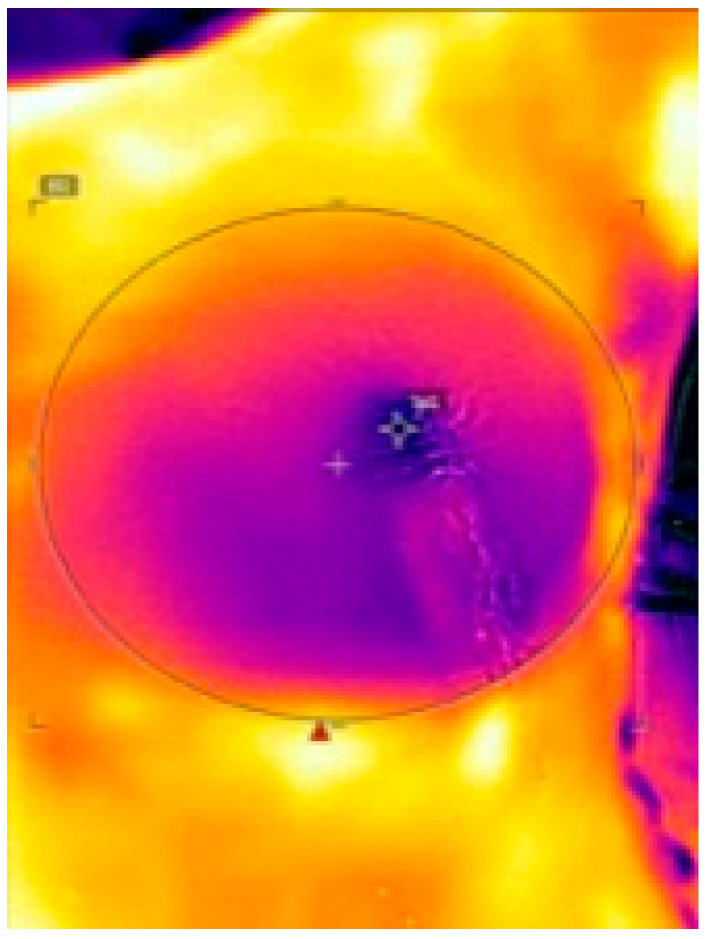
Intraoperative thermal imaging via FLIR ONE. The ROI (black circle) displays the subcutaneous mastectomy area, within the temperature was measured. Darker colors (purple, blue) indicate lower surface temperature while brighter colors (orange, yellow) indicate higher temperature.

**Figure 2 jpm-12-00740-f002:**
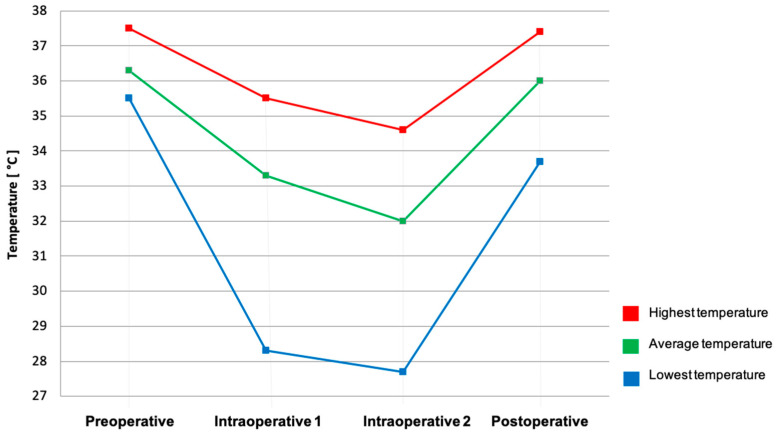
Surface temperature changes during the procedure of NSM/SSM and immediate reconstruction with implants. Data are presented as mean values [°C].

**Figure 3 jpm-12-00740-f003:**
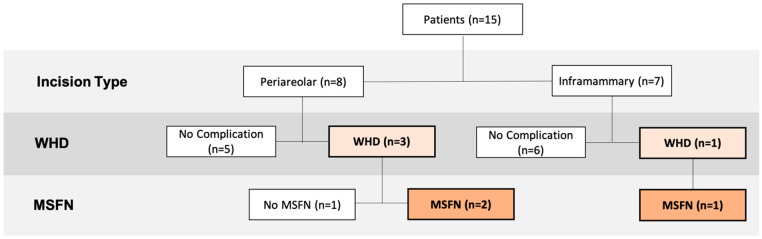
Incision types and postoperative complications. Abbreviations: Mastectomy-skin-flap necrosis (MSFN); wound-healing disorder (WHD).

**Table 1 jpm-12-00740-t001:** Surface temperatures at different measurement points. Data are presented as average values [°C] and standard deviations. An asterisk indicates statistical significance.

	Highest T. [°C]	Lowest T. [°C]	Average T. [°C]
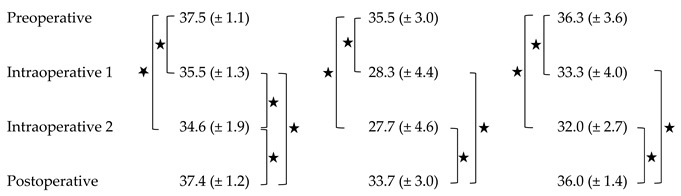			

Abbreviations: Temperature (T.).

**Table 2 jpm-12-00740-t002:** Surface temperatures in the no-complication and the complication groups. Data are presented as average values and standard deviations. An asterisk indicates statistical significance (*p* < 0.05).

	Highest T. [°C]		Lowest T. [°C]		Average T. [°C]	
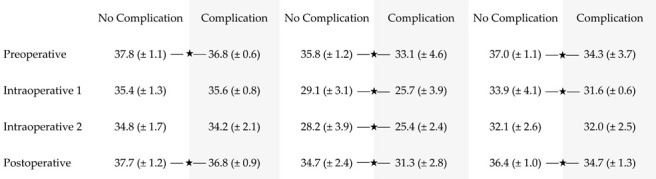						

Abbreviations: Temperature (T.).

**Table 3 jpm-12-00740-t003:** Patient-, surgery- and environmental-related factors. Data are presented as average values and standard deviations. An asterisk indicates statistical significance between the groups.

		No Complication	Complication
Patients (*n* = 15)		11 (73.3%)	4 (26.7%)
Age [years]		47.6 (±10.6)	46.3 (±1.5)
BMI [kg/m^2^]		25.3 (±3.0)	28.1 (±0.6)
Body temperature [°C]	★	36.0 (±0.1)	35.6 (±0.4)
Surgery duration [min]	★	142 (±0.43)	178 (±42)
Implant size [cm^3^]	★	320.5 (±42.5)	370 (±43.3)
Room temperature [°C]	★	23.0 (±0.2)	21.2 (±1.2)

Abbreviations: Body mass index (BMI).

## Data Availability

The authors confirm that the data supporting the findings of this study are available within the article.

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
