# Peer review of "Assessment of Mastectomy Skin Flaps for Immediate Reconstruction with Implants via Thermal Imaging—A Suitable, Personalized Approach?"

_jpm, 2022, doi:10.3390/jpm12050740_

Round 1
Reviewer 1 Report
Extensive syntax and grammar editing should be accomplished. I suggest a careful revision of the entire manuscript to improve readability.
References should be revised. Authors names are missing.
Inclusion criteria should be clearly stated
How many patient underwent radio/chemotherapy?
Follow-up length should be stated
Line 203 “Statistically significant lower mean body temperatures (35.6°C (SD ± 0.4°C), p = 0.012) were noted in the complication group compared to the no-complication group (36.0 °C (SD ± 0.1°C)”. Is this a preoperative, intraoperative or post operative data?
Tabel 3. P-values shoud be added
How many patients underwent NSM and SSM and risk reducing mastectomy?
Results line 133-135 should be revised. SD is associated with a rate.
Author Response
Reviewer #1:
1) In response to the comment of reviewer #1 we have performed an extensive grammar editing throughout the entire manuscript to improve readability.
2) Thank you very much for your comment on the reference list, which has been revised: authors of references No. 21,23 and 28 were added.
3) The reviewer noted that the inclusion and exclusion criteria should be better defined, according to this comment, we changed the following section:
Line 90-95:
Fifteen female (age range 18 - 80 years) healthy, non-pregnant study participants with a history of BC, carcinoma in situ on one or both breasts and or known genetic BRCA I or BRCA II mutation undergoing planned NSM or SSM for a therapeutic or risk reduction indication for breast cancer and following IBR with implants were enrolled. Exclusion criteria were defined as follows: BC diagnosis with the contraindication for NSM or SSM (e.g. inflammatory carcinoma), Diabetes mellitus Type 1 and 2, nicotine abuse, inability to fully comprehend study procedures or to provide written informed consent.
4) The question, how many patients underwent radio or chemotherapy, as well as how many NSM, SSM and risk reducing mastectomies were performed, was raised. Each patient enrolled underwent chemotherapy prior to surgery, however, no previously performed radiotherapy was noted. Unfortunately, we were only able to include one patient with a risk reduction mastectomy, whereby no postoperative complications were observed. Furthermore, NSM was performed in 6 and SSM in 7 patients. We have noted this information in the results section as follows:
Line 133-137:
Overall, 15 patients with a mean age of 44.1 years (SD ± 9.2) and a mean BMI of 25.9 (SD ± 2.46) were included in this investigation. All patients enrolled underwent neoadjuvant chemotherapy, no patient underwent radiotherapy prior to surgery. Risk reduction mastectomy was performed in one, NSM in six and SSM in seven patients. Mean surgery duration was 2 hours 32 minutes (SD ± 108 min). 8 (53.33%) periareolar and 7 (46.67%) inframammary incisions were performed.
Furthermore, a paragraph regarding the correlation of the NSM and SSM technique and the incidence of postoperative complications was added in the results section as follows:
Line 179-180:
No statistically significant difference between incision type (p = 0.36) and NSM or SSM technique (p = 0.29) was observed.
5) The reviewer stated, that the follow-up length should be stated. Since the present project primary focused on immediate postoperative complications due to underperfusion of the mastectomy skin flap, its’ analysis via thermal imaging was performed up to 24 hour postoperative. However, the follow-up visits were scheduled up to 6 weeks postoperative, which was stated as follows:
Line 98-99:
Postoperative follow-up visits were uniformly scheduled at 1, 2 and 6 weeks postoperative.
6) Line 195: The values in this sentence are mistakenly not fully specified: these values correspond to the average of all temperature values in the complication and no-complication group before and during the procedure, until 24 hours postoperative. To clarify this sentence, it was changed to the following:
Line 195-196:
Statistically significant lower mean body temperatures during the entire procedure (35.6°C (SD ± 0.4°C), p = 0.012) were found in the complication group when compared to the no-complication group (36.0 °C (SD ± 0.1°C) .
7) The reviewer suggested to add p-values within Table 3, where statistical significance is indicated with an asterisk. However, for overview reasons and better readability, we do not want to specify the values since they are clearly stated within the text.
8) Results in line 136 were updated as follows:
Line 135:
8 (53.33%) periareolar and 7 (46.67%) inframammary incisions were performed.

Reviewer 2 Report
As mentioned by the authors the statistical results are questionable because of the very small number of patients especially concerning the influence of the patient-related factors.
The measured temperature differences seem to be sufficiently significant at least to be taken as a relevant indication for a minor perfusion.
The advantage of the temperature measurement as an secondary perfusion parameter seems to be the convenience of the used FLIR-system in comparison with other methods measuring the perfusion more directly.
Resultats and discussion
- In row 192 -194 the room temperature for the complication group (23,0°) should be lower than for the non-complication group (21,2°)?
- Has been analysed a possible correlation between the temperature measurement and room temperature by varying the room temperature?
The individual risk factors normally cannot be influenced, but the OP strategy and the frequency of perfusion testing can be adapted to enable timely interventions, as mentioned in the discussion.
Following the logic of the manuscript, an individual adaption of the actually globally fixed threshold of 26° should be investigated in further studies.
Author Response
Reviewer #2:
1) Thank you very much for pointing out the limitation of the present project due to the small sample size. We tried to put more emphasis on the statement as follows:
Line 280-285:
This study is limited due to the small sample size, hence results and patient-related factors in particular may not be fully representative and transferable. Despite some significances that were shown, the small sample size might be the reason why further significances could not be reached, but tendencies were shown.
2) The reviewer stated, that in row 192 -194 the room temperature for the complication group (23,0°) should be lower than for the non-complication group (21,2°). We have found a number error in this case, which has been corrected as follows:
Line 185-186:
Furthermore, statistically significant lower average room temperatures (21.2°C (SD ± 0.2 °C), p= < 0.001) were found in the complication group in comparison to the no-complication group (23.0°C (SD ± 1.2°C)).
3) The question, whether a possible correlation between the temperature measurement and room temperature by varying the room temperature has been investigated. However, we did not vary the room temperature during the procedure, hence did not investigate any correlation top the skin surface temperature. Our results might have been biased by variation of the room temperature due to the small sample size. Since we found lower average room temperatures in the complication group, the question of a correlation with the skin surface temperature should definitely be addressed in future, lager cohort studies.

Round 2
Reviewer 1 Report
The authors revised the manuscript; however, language is still far from optimal (i.e. line 54 “most commonly reason”; line 77-79 and line 90-92 syntax should be checked)
Inclusion criteria should be clearly stated as per exclusion criteria. A flow chart could be useful to assess how many patients were assessed for eligibility and how many were enrolled.
Check the unit of measure (i.e. BMI)
Line 135. “Risk reduction mastectomy was performed in one, NSM in six and SSM in seven patients. If a patient underwent surgery on both breast, this should be clearly stated“ 15 patients were enrolled in this study. Please revise
Conclusions section should be reduced, so the reader could get the focus of this study.
Author Response
Reviewer #1:
1) In response to the comment of reviewer #1 the manuscript was proved again by a native speaker who performed moderate grammar editing throughout the entire manuscript. In particular, the following issues pointed out by the reviewer have been revised:
Line 54-55
Breast cancer (BC) ranks top in malignancies among females worldwide and represents the most common reason for death by cancer as well (1).
Line 77-79:
The present study evaluated the feasibility of assessing the individual mastectomy skin flap perfusion via the thermal imaging device FLIR ONE in patients undergoing NSM or SSM and following IBR with implants.
2) The reviewer asked for inclusion criteria to be as clearly stated as per exclusion criteria. We have revised this paragraph as follows:
Line 90-92:
Fifteen female (age range 18 - 80 years) healthy, non-pregnant study participants were enrolled. Inclusion criteria were defined as follows: history of BC, carcinoma in situ on one or both breasts and/or known genetic BRCA I or BRCA II mutation with planned NSM or SSM and following IBR with definite implants.
3) The unit of measure (BMI) has been changed in line 132 and Table 3.
4) The reviewer pointed out, that risk reduction mastectomy was performed in one patient and a clear statement, that surgery was performed on both breasts in this patient should be included. It is correct, that this patient underwent risk reduction mastectomy on both sides, however, IBR was performed using a definite implant on one side and a minimally prefilled tissue expander on the other. To avoid bias and meet our inclusion criteria, we assessed skin flap perfusion on the side with IBR with a definite implant exclusively.
To clarify this issue, we added the following phrase:
Line 135 -137:
In the patient undergoing risk reduction mastectomy, skin flap perfusion was assessed one side exclusively, since a minimally prefilled tissue expander was used on the other side.
5) The conclusion was shortened to ensure better readability and focus on the key message of our manuscript as follows:
Line 294-300:
xWHD and MSFN due to compromised perfusion patterns of the remaining skin flap occur more commonly than appreciated, leading to numerous challenges. Hypoperfusion of the remaining mastectomy skin flap as a major factor contributing to WHD and MSFN is readily and accurately examined via the novel and personalized approach of thermal imaging with the FLIR device. Skin areas with a temperature lower than 26°C are highly likely to develop subsequent WHD and MSFN and may require early intervention to avoid or reduce the incidence of MSFN. Ultimately, careful and diligent individual preoperative planning and intraoperative monitoring may contribute to a reduced incidence of WHD converting to MSFN.
